# Angiopoietin-2 and Angiopoietin-like Proteins with a Prospective Role in Predicting Diabetic Nephropathy

**DOI:** 10.3390/biomedicines12050949

**Published:** 2024-04-24

**Authors:** Eman Alshawaf, Mohamed Abu-Farha, Anwar Mohammad, Sriraman Devarajan, Irina Al-Khairi, Preethi Cherian, Hamad Ali, Hawra Al-Matrouk, Fahd Al-Mulla, Abdulnabi Al Attar, Jehad Abubaker

**Affiliations:** 1Department of Biochemistry and Molecular Biology, Dasman Diabetes Institute, Dasman 15462, Kuwait; eman.alshawaf@dasmaninstitute.org (E.A.); mohamed.abufarha@dasmaninstitute.org (M.A.-F.); anwar.mohammad@dasmaninstitute.org (A.M.); irina.alkhairi@dasmaninstitute.org (I.A.-K.); preethi.cherian@dasmaninstitute.org (P.C.); 2Diabetology Unit, Amiri Hospital, Dasman Diabetes Institute, Dasman 15462, Kuwait; 3National Dasman Diabetes Biobank, Dasman Diabetes Institute, Dasman 15462, Kuwait; sriraman.devarajan@dasmaninstitute.org; 4Functional Genomic Unit, Dasman Diabetes Institute, Dasman 15462, Kuwait; hamad.ali@ku.edu.kw (H.A.); fahd.almulla@dasmaninstitute.org (F.A.-M.); 5Department of Medical Laboratory Sciences, Faculty of Allied Health Sciences, Health Sciences Center, Kuwait University, Kuwait 15462, Kuwait; 6Medical Department, Amiri Hospital, Ministry of Health, Kuwait 15462, Kuwait; hawra.h.almatrouk@gmail.com; 7Department of Translational Research, Dasman Diabetes Institute, Dasman 15462, Kuwait

**Keywords:** Ang1, Ang2, ANGPTL8, ANGPTL4, DN

## Abstract

Angiopoietins are crucial growth factors for maintaining a healthy, functional endothelium. Patients with type 2 diabetes (T2D) exhibit significant levels of angiogenic markers, particularly Angiopoietin-2, which compromises endothelial integrity and is connected to symptoms of endothelial injury and failure. This report examines the levels of circulating angiopoietins in people with T2D and diabetic nephropathy (DN) and explores its link with ANGPTL proteins. We quantified circulating ANGPTL3, ANGPTL4, ANGPTL8, Ang1, and Ang2 in the fasting plasma of 117 Kuwaiti participants, of which 50 had T2D and 67 participants had DN. The Ang2 levels increased with DN (4.34 ± 0.32 ng/mL) compared with T2D (3.42 ± 0.29 ng/mL). This increase correlated with clinical parameters including the albumin-to-creatinine ratio (ACR) (r = 0.244, *p* = 0.047), eGFR (r = −0.282, *p* = 0.021), and SBP (r = −0.28, *p* = 0.024). Furthermore, Ang2 correlated positively to both ANGPTL4 (r = 0.541, *p* < 0.001) and ANGPTL8 (r = 0.41, *p* = 0.001). Multiple regression analysis presented elevated ANGPTL8 and ACRs as predictors for Ang2’s increase in people with DN. In people with T2D, ANGPTL4 positively predicted an Ang2 increase. The area under the curve (AUC) in receiver operating characteristic (ROC) analysis of the combination of Ang2 and ANGPTL8 was 0.77 with 80.7% specificity. In conclusion, significantly elevated Ang2 in people with DN correlated with clinical markers such as the ACR, eGFR, and SBP, ANGPTL4, and ANGPTL8 levels. Collectively, this study highlights a close association between Ang2 and ANGPTL8 in a population with DN, suggesting them as DN risk predictors.

## 1. Introduction

Diabetes mellitus is a significant global public health challenge [1]. Epidemiological studies indicate that the proportion of people affected by diabetes was approximately 8.8% in 2015, and this percentage is projected to increase to 10.4% by 2024. An analysis released in 2020, utilizing data from 2014, determined that the prevalence of diabetes in Kuwait was projected to reach 21.8% [2]. Diabetic nephropathy (DN) is a significant microvascular complication and the primary cause of mortality among individuals with diabetes [3]. According to the tenth report of the International Diabetes Federation (IDF), approximately 7.8% of people in Kuwait are diagnosed with DN as a consequence of T2D [4]. Thus, DN imposes a substantial cost on individuals affected by the condition and on healthcare systems.

The angiopoietin (Ang)/Tie ligand-receptor system is of paramount importance in the maintenance of endothelial integrity and vascular development [5]. Alterations in the Ang/Tie system have been suggested to contribute to the advancement of kidney injuries in individuals with DN [6]. Dysregulation of the angiopoietin balance, specifically Ang1 and Ang2, and their interaction with Tie receptors (Tie1 and Tie2) can have detrimental effects on endothelial function and vascular stability in the renal microvasculature [6,7]. Disruption in Tie receptor signaling and an imbalance between Ang1 and Ang2 collectively contribute to heightened vascular permeability, inflammation, and oxidative stress. These various factors have the potential to induce glomerular dysfunction and structural alterations in the kidneys, thereby playing a role in the advancement of diabetic nephropathy [6,7]. In general, the pathogenesis of diabetic nephropathy is characterized by an intricate interplay of metabolic and hemodynamic factors. The role of angiogenesis in the development of DN is well acknowledged [8], and consequently, growth factors associated with angiogenesis, such as Ang1 and Ang2, will have a substantial impact on the development of DN. Patients with T2D were found to have elevated levels of circulating Ang2 [9], which led to microalbuminuria [10]. The rise in Ang2 levels acted as an independent predictor of microalbuminuria in patients with T2D [10] and was reported in connection with a rapid decrease in kidney function [11]. Significantly elevated Ang2 levels were also positively linked with the progression of albuminuria in people with DN [8,12,13].

Angiopoietin-like proteins (ANGPTLs) are a family of proteins structurally similar to angiopoietins. To date, eight ANGPTLs have been discovered, namely ANGPTL1–ANGPTL8. ANGPTLs, particularly ANGPTL3, ANGPTL4, and ANGPTL8, play pivotal roles in regulating lipid metabolism and energy homeostasis, with emerging implications for diabetic nephropathy. ANGPTL3 and ANGPTL4 are known inhibitors of lipoprotein lipase (LPL), a key enzyme involved in triglyceride metabolism. By inhibiting LPL activity, these proteins modulate lipid clearance from the circulation, resulting in alterations in plasma lipid levels. ANGPTL8, often acting in conjunction with ANGPTL3, also regulates LPL activity and lipid metabolism, albeit with some distinct roles in glucose homeostasis and insulin sensitivity [14]. Dysregulation of these ANGPTLs has been implicated in the pathogenesis of metabolic disorders, including obesity, dyslipidemia, and type 2 diabetes mellitus (T2DM), all of which are risk factors for diabetic nephropathy [15]. ANGPTL8 is a hepatic protein that is produced in the liver and adipose tissue. Despite its lack of direct involvement in angiogenesis, ANGPTL8 has been identified as a significant contributor to regulating metabolism processes by adjusting glucose and lipid levels, in addition to its crucial role in maintaining lipid balance [14]. In addition to its association with T2D [16,17], ANGPTL8 is linked to diabetes complications and other concomitant disorders such as DN [15]. Chen et al. reported that ANGPTL8 is considerably higher in T2D patients with various stages of albuminuria [18]. Having said that, ANGPTL8 is thought to be involved in DN, and further research into its mechanism in T2D and DN is needed. The collective involvement of Ang proteins and ANGPTL8 in vascular biology and metabolic equilibrium renders them highly promising targets for diagnostic and therapeutic interventions in conditions such as metabolic syndromes and cardiovascular disorders. This study examines the correlation between Ang2, ANGPTL8, and DN to assess the potential predictive significance of ANGPTL8 and Ang2 in DN.

## 2. Materials and Methods

### 2.1. Study Population

A total of 117 participants joined this study, and the participants were segregated into two groups. Throughout the report, people diagnosed with diabetes are referred to as T2D, and those diagnosed with diabetes and nephropathy are designated as DN. The participants were body mass index (BMI) and age matched among the individuals with T2D (n = 50) and people with T2D and nephropathy (DN group, n = 67). All participants provided written consent at the Dasman Diabetes Institute (Dasman, Kuwait) to be enrolled in the study. Ethical approval for this study was granted by the Ethical Review Board of the Dasman Diabetes Institute (DDI), abiding by the ethical guidelines outlined in the Declaration of Helsinki.

### 2.2. Anthropometric and Biochemical Measurements

Blood pressure was measured with an Omron HEM-907XL digital sphygmomanometer, and the presented values are the average of three consecutive readings. Plasma was extracted from fasting blood samples collected in vacutainer-EDTA tubes (centrifugation for 10 min at 400× *g*). The plasma samples were aliquoted and stored at −80 °C for further testing. The fasting blood glucose (FBG), serum total cholesterol (TC), low-density lipoprotein (LDL-C), high-density lipoprotein (HDL-C) and triglycerides (TG) were measured with a Siemens Dimension RXL chemical analyzer (Diamond Diagnostics, Holliston, MA, USA). Quantification of albumin and creatinine in the urine samples was performed with a CLINITEK Novus Automated Urine Chemical Analyzer (Siemens Healthineers, Erlangen, Germany).

### 2.3. Quantification of Creatinine and Urinary Protein

The urinary and serum creatinine levels were measured with a VITROS 250 automatic analyzer (New York City, NY, USA). Urinary proteins were quantified using a Coomassie Plus protein assay kit according to the manufacturer’s protocol (pierce, Rockford, IL, USA). The estimated glomerular filtration rate (eGFR) was calculated using a modification of the Diet in Renal Disease study equation.

### 2.4. ANGPTL3, 4, and 8 Enzyme-Linked Immunosorbent Assays (ELISAs)

The levels of plasma in the ANGPTL3, ANGPTL4 and ANGPTL8 proteins were quantified using a Magnetic Luminex Assay kit (R&D Systems Europe, Ltd., Abingdon, UK), following the manufacturer’s protocol. Repeated freeze-thaw cycles of the plasma samples were avoided, and all samples were thawed on ice before assaying. Cross-reactivity with other proteins was not significant.

### 2.5. Quantification of Ang1 and Ang2

The levels of Ang1 and Ang2 were determined by the Magnetic Luminex Assay kit (R&D Systems Europe, Ltd., Abingdon, UK), following the manufacturer’s protocol.

### 2.6. Statistical Analysis

An unpaired student’s *t*-test was used to compare the two study groups (i.e., people with T2D and people with DN) to determine statistical significance. The correlation analysis between Ang1, Ang2, and various parameters was estimated by Pearson’s correlation coefficient. A stepwise multiple linear regression model was performed to identify the parameters independently associated with Ang1 and Ang2. The diagnostic strength of Ang2 as well as ANGPTL8 as biomarkers for DN was calculated through area under the receiver operating characteristic (ROC) curve analysis. Similarly, the diagnostic strength of Ang2 with Ang1 as biomarkers for DN was calculated by ROC curve analysis. All data are presented as the mean ± SEM, with a *p* value < 0.05 indicating significance. All statistical analysis was performed using Graphpad Prism Software version 9 (La Jolla, CA, USA) and SPSS for Windows version 25.0 (IBM SPSS Inc., New York City, NY, USA).

## 3. Results

Our study involved a total of 117 Arab participants recruited from the population of the state of Kuwait. Descriptions of the population demographic and various clinical parameters are detailed in Table 1. The participants were segregated into two main study groups: people diagnosed with T2D (n = 50) and people diagnosed with diabetes and nephropathy (DN = 67). Diagnosis of type 2 diabetes was determined by having an elevated fasting blood sugar (i.e., ≥7 mmol/L) and glycated hemoglobin (A1C) (i.e., ≥6.5%) test results. Diagnosis of diabetic nephropathy was determined with the presence of a severely increased albumin-to-creatinine ratio (ACR) (i.e., >30 mg/g).

### 3.1. Circulating Angiopoietins 1 and 2 Are Elevated in People with DN

In this study, we detected an increase in the circulating levels of Ang1 and Ang2 in people with DN compared to people with T2D. People with DN showed elevated levels of Ang1, but the increase in circulating Ang1 was statistically insignificant compared with the people with T2D (Figure 1A). On the other hand, there was a significant increase in the levels of Ang2 in the people with DN (*p* = 0.034, Figure 1B) compared with those with T2D (Table 1). In addition to the angiopoietins, our data showed an increase in some angiopoietin-like proteins in the people with DN. Both ANGPTL4 (*p* = 0.029, Figure 1C) and ANGPTL8 (*p* ≤ 0.001, Figure 1D) showed a significant increase in their circulating levels compared with people with T2D (Table 1).

### 3.2. Increased Ang2 Correlated with Clinical Parameters of DN

To further examine the significance of the angiopoietins’ elevation, for both Ang1 and Ang2, under the conditions of diabetes and kidney disease, we employed Pearson’s correlation analysis to explore the link between the clinical parameters indicative of DN and angiopoietin proteins. When performing the correlation analysis, we found a significant negative correlation between the elevation of Ang1 in the plasma and albumin (r = −0.24, *p* = 0.045) and a positive correlation with both microalbumin (r = 0.35, *p* = 0.003) and ANGPTL3 (r = 0.25, *p* = 0.03) in people with DN (Table 2). By implementing Pearson’s analysis, we identified a correlation between Ang2 and several clinical parameters in people with DN and T2D (Figure 2). Here, we present a negative correlation between increased Ang2 levels and eGFR in people with DN (r = −0.28, *p* = 0.021; Figure 2A) and people with T2D (r = −0.307, *p* = 0.034), while elevated Ang2 levels showed a negative correlation with systolic blood pressure (r = −0.28, *p* = 0.024) only in people with DN (Table 3). Furthermore, increased levels of Ang2 showed positive correlations with elevated levels of ACR (r = 0.24, *p* 0.047; Figure 2B), ANGPTL4 (r = 0.54, *p* ≤ 0.001; Figure 2C), and ANGPTL8 (r = 0.41, *p* = 0.001; Figure 2D) in people with DN (Table 3). Our analysis also showed that increased Ang2 levels in people with T2D was positively correlated with the levels of serum creatinine (r = 0.3, *p* = 0.036), BUN (r = 0.32, *p* = 0.025), and ANGPTL4 (r = 0.55, *p* ≤ 0.001) (Table 3).

### 3.3. Predictive Analysis Suggests a Potential Ang2-ANGPTL8 Link with DN

We implemented multiple stepwise regression analysis with a set of predictors to gain further insight into the potential connection between Ang2 and the biochemical diagnostic parameters (Table 4). According to our model, elevated ANGPTL8 levels and ACRs are both predictive markers with a significant positive regression weight for the elevation of Ang2 in people with DN (Table 4). These markers showed an independent correlation with Ang2 and ANGPTL8 levels (F_1,58_ = 3.171, *p* < 0.001, and r^2^ = 34%) and ACRs (F_1,58_ = 2.611, *p* = 0.031, and r^2^ = 30%), while SBP acted as a negative predictor of increased Ang2 levels in people with DN (F_1,58_ = 4.102, *p* = 0.039, and r^2^ = 32%) (Table 4). Collectively, these markers independently correlated with the increase in the Ang2 level. On the other hand, the people with T2D presented ANGPTL4 (β = 0.552, *p* < 0.001; Table 4) as a positive independent predictor for increased Ang2 levels (F_1,47_ = 11.008, *p* < 0.001, and r^2^ = 30%). To sum up, our data revealed an independent correlation between SBP, the ACR and ANGPTL8 with Ang2 in people with DN, thus highlighting these markers as significant predictors, whereby Ang2 is a dependent variable in people with diabetic nephropathy.

Additionally, we performed ROC curve analysis to identify the cut-off value of Ang2 and ANGPTL8 and to further evaluate the predictive accuracy of these biomarkers for people with DN (Figure 3). Our analysis showed that the area under the curve (AUC) (95% CI) was 0.74 (0.66–0.81, *p* < 0.001) for Ang2, 0.79 (0.70–0.89, *p* < 0.001) for ANGPTL8, and 0.77 (0.70–0.85, *p* < 0.001) for the combination of Ang2-ANGPTL8. The optimal cut-off value for predicting DN with Ang2 was higher than 1426.78 ng/mL with 93% sensitivity and a specificity of 86%. The optimal cut-off value for ANGPTL8 as a predictive marker for DN was higher than 1135.75 ng/mL, with 93.9% sensitivity and a specificity of 81.1%. Additionally, the optimal cut-off value for the combination of Ang2 and ANGPTL8 was higher than 2681.08 ng/mL, with 94.1% sensitivity and a specificity of 80.7%. This finding indicates that ANGPTL8 may possess superior diagnostic accuracy compared with Ang2 and the combined biomarkers in distinguishing diabetic nephropathy from other conditions or outcomes.

ROC curve analysis was also performed to identify the cut-off value of Ang2 and Ang1 as potential predictive biomarkers for people with DN in the study population (Figure 4). Our analysis showed that the AUC (%95 CI) was 0.72 (0.65–0.80, *p* < 0.001) for Ang1, 0.74 (0.66–0.81, *p* < 0.001) for Ang2, and 0.81 (0.74–0.88, *p* < 0.001) for the combination of Ang1-Ang2. The identified cut-off value for predicting DN with Ang1 was above 1590.92 ng/mL, with 94% sensitivity and a specificity of 77%. As for Ang2, the optimal cut-off value for predicting DN was above 1426.78 ng/mL, with 93% sensitivity and a specificity of 86%. The combination of Ang1 and Ang2 demonstrated an optimal cut-off value exceeding 1518.91 ng/mL, with a sensitivity of 92% and a specificity of 89%.

## 4. Discussion

In this study, we verified that individuals with DN have elevated levels of circulating Ang2 and ANGPTL8 compared with those with T2D. Our analysis revealed for the first time a positive correlation between the elevation in Ang2 and rising ANGPTL8 and ANGPTL4 levels in patients with DN. Additionally, the ROC curve analysis indicated the sensitivity and specificity of using Ang2 in combination with ANGPTL8 as a diagnostic tool for people with DN, suggesting their potential as significant predictors for nephropathy in patients with T2D.

In agreement with previous studies, we report that blood levels of Ang2 [11,12] and ANGPTL8 [19] were higher in the people with DN. A noteworthy finding in the current report was that the rise in Ang2 exhibited a direct and strong link with ANGPTL8 and the clinical parameters of DN, such as the ACR and eGFR. Ang2 is a growth factor belonging to the angiopoietin/tyrosine kinase signaling pathway that is upregulated in animal models of kidney disease [20,21,22,23] and in diabetic nephropathy [24,25,26].

Angiopoietins are a group of vascular growth factors that have several physiological roles associated with vascular development and repair. The actions of Angs are facilitated by endothelial Tie receptor tyrosine kinases via the Ang-Tie signaling pathway, which helps with angiogenesis in health and disease (i.e., vascular diseases, systemic inflammation, and cancers). Angiopoietins are present in two isoforms, Ang1 and Ang2, as they contribute to regulating vascular homeostasis. Ang1 and Ang2 share significant amino acid similarities (≥65%) (Appendix A), whereby the fibrinogen-like domains (FLDs) of Ang1 or Ang2 possess the same structural fold (Appendix A) [27]. The FLD domains of Ang1 or Ang2 bind to the Ig2 domain of the Tie2 protein (Appendix A). The Ang1/Tie2 interface is similar to the Ang2/Tie2 interface (Appendix A). Studies have demonstrated that the binding affinity of Ang2 to Tie2 is less than Ang1 and less potent in activating Tie2 [8]. However, when Ang1 and Ang2 are added simultaneously, Ang1-Tie2 phosphorylation is inhibited, diminishing the protective effect of Ang1 [8]. Functionally, Ang1 plays a critical protective role in endothelial cells (ECs) by initiating an anti-inflammatory response and inducing vessel wall stabilization. The activity of Ang1 is attained via binding and inducing autophosphorylation, thus activating Tie2. On the contrary, Ang2 acts as a natural antagonist of a pro-inflammatory nature toward Ang1 and its Tie2 receptor [28]. The Ang-Tie signaling pathway has critical importance, particularly in the development and maturation of the kidney, as well as acute and chronic kidney diseases like DN, lupus nephropathy, hemolytic uremic syndrome, end-stage renal diseases, and renal cell carcinoma [6]. Ang2 is mainly produced by endothelial cilia [29] and stored in the Weibel–Palade bodies within ECs [30]. The occurrence of ischemia, hypoxia, or inflammation induces the release of Ang2 into circulation [31], where it competes with Ang1 to inhibit the phosphorylation and activation of the Tie2 receptor. Consequently, Ang2 facilitates permeability of the endothelium, which leads to destabilization of the blood vessel wall [32].

Our data presents a significant rise in the plasma levels of Ang2 in people with DN, which is in agreement with Tsai et al., who found a significant connection between elevated Ang2 levels and an increased risk of poor renal outcomes in patients with diabetic nephropathy [11]. However, the significance of Ang2 and its role in kidney pathophysiology is still obscure. A growing body of research indicates that the elevation of Ang2 has negative effects on renal physiology and function. Additionally, increased Ang2 expression impacts podocytes through paracrine signaling, leading to glomerular EC destabilization that consequently deteriorates the function of the glomerular filtration barrier [33]. Furthermore, normoalbuminuric patients with T2D presented elevated levels of Ang2 in their blood and urine [10,34]. Collectively, this compiling evidence highlights the importance of Ang2 as an early marker of tubular damage before the appearance of clinical symptoms like microalbuminuria.

ANGPTL8 moderates plasma triglyceride levels by blocking lipoprotein lipases. Multiple studies have indicated an increase in circulating ANGPTL8 in individuals with T2D [35,36,37,38,39,40,41]. Consistent with the findings showing elevated ANGPTL8 levels in individuals with T2D, we previously reported a threefold rise in circulating ANGPTL8 in people with T2D compared with healthy individuals. Additionally, we have shown that the circulating ANGPTL8 level is higher in people with DN compared with people with T2D. The increase in ANGPTL8 correlated with the clinical parameters of nephropathy in the same population with DN [42]. This rise aligns with the reported link between varying levels of ANGPTL8 and different phases of albuminuria [18,43]. ANGPTL8 was proposed as a new regulator in DN development and as a new predictive risk indicator for all-cause death in individuals with T2D [41], and it was highlighted as a predictive marker for diabetic complications, specifically DN and declining kidney function [19,42]. In this report, individuals diagnosed with DN exhibited microalbuminuria (460.34 ± 169.37 mg/day, *p* = 0.001), which is consistent with the documented association between elevated circulating ANGPTL8 levels and albuminuria [18]. The increase in ANGPTL8 synthesis was ascribed to albumin loss, a condition that leads to insulin resistance and heightened insulin requirements in individuals with T2D and albuminuria.

In individuals with diabetic nephropathy, our data presents a notable rise in the levels of circulating Ang2, ANGPTL8, and ANGPTL4. Among these, Ang2 showed significant associations with certain clinical markers of DN. The Ang2 levels increased in conjunction with higher levels of ANGPTL8 and ANGPTL4 and the ACR but showed a negative relationship with the eGFR and systolic blood pressure, suggesting possible involvement in diabetic nephropathy. The proven association between Ang2 and nephropathy [6,7,9,11], as well as the link between ANGPTL8 and renal dysregulation [18,41,42], and the presence of a positive correlation between Ang2 and ANGPTL8 all point to a possible interplay between these proteins contributing to DN’s progression or severity. The detected rise in circulating ANGPTL4 is in agreement with a prior report [44], and it exhibited a significant positive correlation with Ang2 in both people with T2D and DN. This implicated a potential link between the two markers in T2D, which was emphasized by the significant multiple regression analysis for ANGPTL4 (β = 0.552, *p* < 0.001; Table 4). Although the Ang2-ANGPTL4 link is currently obscure, multiple regression data (Table 4) highlights the importance of changes in ANGPTL4 in T2D as a predictive marker for changes in Ang2. On the other hand, the larger AUC for ANGPTL8 underscores its potential as a robust biomarker for diabetic nephropathy, offering promise for improved diagnostic precision and patient stratification in clinical practice. Finally, the ROC analysis of the amalgamation of Ang2 and ANGPTL8 accentuates the significance of these proteins as prospective indicators for the progression into a state of nephropathy in people with T2D. Future research is needed to explain the relationship between Ang2 and ANGPTLs, particularly ANGPTL8 and ANGPTL4, as well as the mechanism by which they contribute to diabetic microvascular problems.

The main limitation of this study is the cross-sectional design of the study, which hindered the ability to determine the biological significance of these proteins and their potential involvement in the pathophysiology of DN and sequentially establish causality. Future research should include prognosis and mechanistic studies to confirm the correlation between Ang2 and ANGPTL(s) in DN. This would also clarify the cause-effect link between DN and these proteins.

## 5. Conclusions

In conclusion, we report a positive correlation between the elevation of Ang2 and increasing levels of ANGPTL8 in individuals with DN and ANGPTL4 in people with T2D, thus suggesting a potential interplay between Ang2 and ANGPTL(s) that influences the development and advancement of DN. This finding suggests that increased levels of Ang2, ANGPTL8, and ANGPTL4 in the bloodstream could be used as predictive indicators to help detect the advancement of early nephropathy in people with T2D. ANGPTL8’s role in lipid regulation suggests that a continuous rise in its levels can cause dyslipidemia and heightened insulin resistance, exacerbating metabolic stress and contributing to the onset of metabolic disorders such as T2D and DN. ROC curve analysis also emphasized the sensitivity and specificity of Ang2 alone and in combination with ANGPTL8 and Ang1 as diagnostic tools for diabetic nephropathy.

## Figures and Tables

**Figure 1 biomedicines-12-00949-f001:**
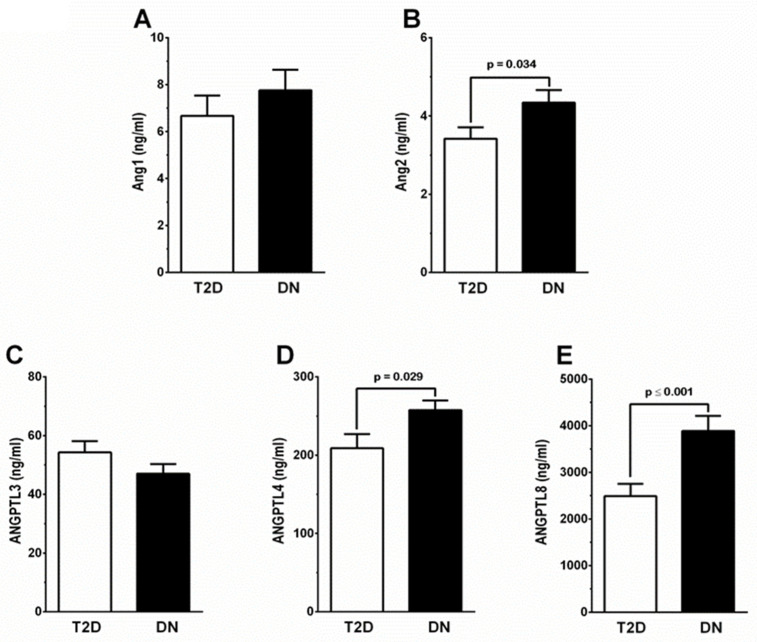
A comparison of circulating levels of Ang1, Ang2, ANGPTL3, ANGPTL4, and ANGPTL8 in people with T2D (white bar, n = 50) and DN (black bar, n = 67). (**A**) Levels of circulating Ang1 were higher in people with DN compared with people with T2D. Difference showed no statistical significance. (**B**) People with DN showed a significant increase in levels of Ang2 (*p* = 0.034) compared with people with T2D. (**C**) ANGPTL3 levels did not show a significant difference between people with T2D and those with DN. (**D**) Circulating levels of ANGPTL4 were significantly higher (*p* = 0.029) in people with DN compared with those with T2D. (**E**) Levels of circulating ANGPTL8 were significantly increased (*p* ≤ 0.001) in people with DN compared with those with T2D.

**Figure 2 biomedicines-12-00949-f002:**
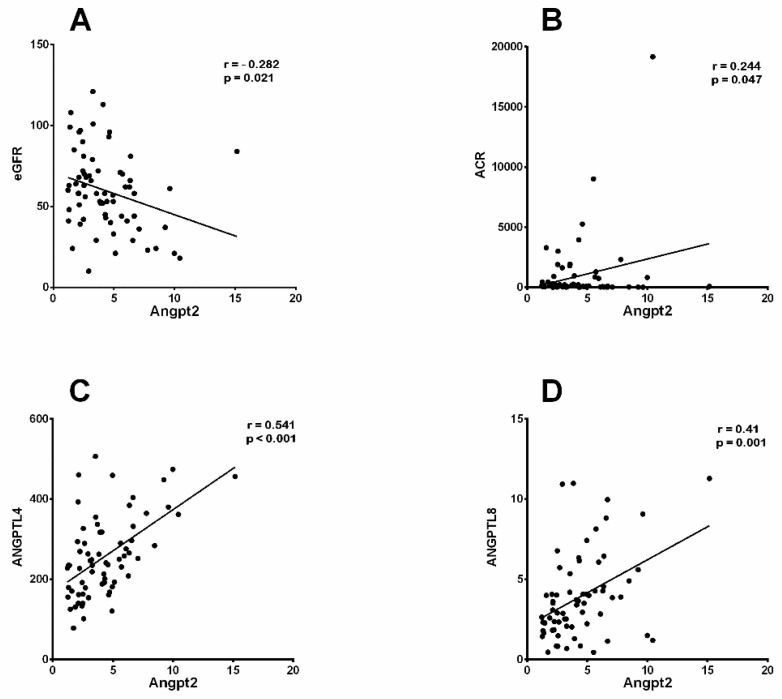
Correlation analysis between Ang2 and parameters associated with DN. Pearson’s correlation coefficient showed a significant (**A**) negative correlation between eGFR and Ang2 (r = −0.282, *p* = 0.021), while elevated Ang2 showed positive correlations with (**B**) ACR (r = 0.244, *p* = 0.047), (**C**) ANGPTL4 (r = 0.541, *p* = 0.001), and (**D**) ANGPTL8 (r = 0.410, *p* = 0.001).

**Figure 3 biomedicines-12-00949-f003:**
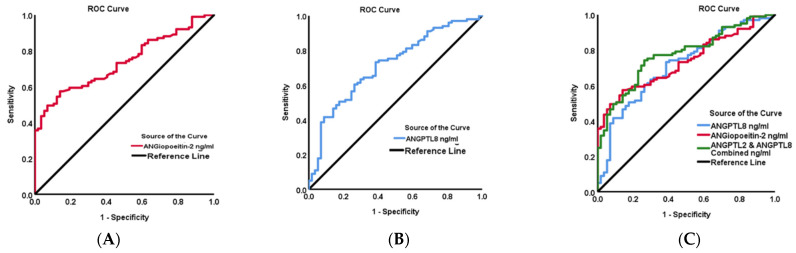
ROC curve analysis performed to identify the cut-off values of Ang2 and ANGPTL8 as biomarkers for DN. (**A**) AUC for Ang2 (0.74 (0.66–0.81) *p* < 0.001). (**B**) AUC for ANGPTL8 (0.79 (0.70–0.89) *p* < 0.001). (**C**) AUC of the combination of Ang2 and ANGPTL8 (0.77 (0.70–0.85) *p* < 0.001).

**Figure 4 biomedicines-12-00949-f004:**
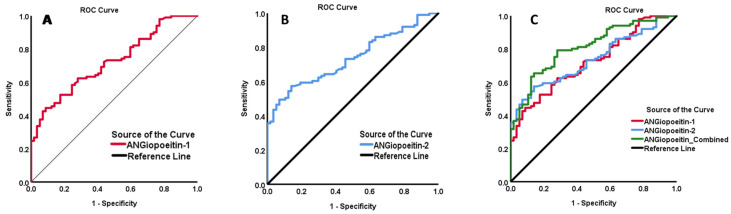
ROC curve analysis performed to identify the cut-off values of Ang1 and Ang2 as biomarkers for DN. (**A**) ANG1 (0.72 (0.65–0.80), *p* < 0.001). (**B**) AUC for Ang2 (0.74 (0.66–0.81) *p* < 0.001). (**C**) AUC of the combination of Ang1 and Ang2 (0.81 (0.74–0.88), *p* < 0.001).

**Table 1 biomedicines-12-00949-t001:** Anthropometric and clinical parameters of people with T2D and diabetic nephropathy.

Parameter	T2D	DN	*p* Value
	N = 50	N = 67	
Gender (M/F)	18/32	45/22	
Age (years)	58.96 ± 1.02	60.09 ± 1.38	0.512
BMI (kg/m^2^)	33.94 ± 0.88	34.23 ± 0.85	0.816
SBP (mmHg)	132.98 ± 3.88	132.03 ± 3.41	0.911
DBP (mmHg)	69.72 ± 2.26	68.78 ± 1.98	0.909
Height (cm)	161.90 ± 1.30	162.04 ± 3.64	0.004
Weight (kg)	88.79 ± 2.35	92.03 ± 2.79	0.055
Fasting Glucose (mmol/L)	8.27 ± 0.36	9.61 ± 0.48	0.028
HbA1C (%)	9.53 ± 1.73	8.09 ± 0.22	0.415
TChol (mmol/L)	4.15 ± 0.13	4.02 ± 0.12	0.472
TG (mmol/L)	1.41 ± 0.16	1.77 ± 0.11	0.066
HDL-C (mmol/L)	1.25 ± 0.05	1.13 ± 0.03	0.067
LDL-C (mmol/L)	2.28 ± 0.11	2.10 ± 0.10	0.203
VLDL (mmol/L)	0.56 ± 0.06	0.71 ± 0.04	0.067
C Peptide (pg/mL)	0.65 ± 0.05	0.77 ± 0.06	0.136
Serum Creatinine (mg/L)	79.42 ± 3.54	118.36 ± 6.57	0.001
eGFR (ml/min/1.73 m^2^)	79.22 ± 3.19	59.70 ± 3.00	0.001
BUN	5.10 ± 0.29	7.53 ± 0.52	0.001
Albumin (mcg/L)	37.94 ± 0.50	37.28 ± 0.42	0.316
Insulin (mU/L)	22.08 ± 3.13	22.22 ± 1.93	0.969
ACR	137.27 ± 69.22	569.94 ± 174.39	0.005
Urine Creatinine (mg/day)	11.38 ± 0.84	10.17 ± 0.91	0.015
Microalbumin (mg/day)	157.58 ± 85.09	460.34 ± 169.37	0.001

Data are mean ± standard error mean. DN = diabetes with nephropathy; SBP = systolic blood pressure; DBP = diastolic blood pressure; ACR = albumin/creatinine ratio.

**Table 2 biomedicines-12-00949-t002:** Pearson’s correlation analysis for Ang1 in study groups T2D and DN.

	Ang1			
Parameters	T2D		DN	
	r	*p*	r	*p*
Age (years)	−0.036	0.808	−0.045	0.715
BMI (kg/m^2^)	0.151	0.301	−0.001	0.994
SBP (mmHg)	0.177	0.228	0.024	0.852
DBP (mmHg)	−0.121	0.414	0.007	0.957
Fasting Glucose (mmol/L)	0.079	0.591	−0.042	0.735
HbA1C (%)	−0.003	0.985	−0.140	0.257
T. Chol (mmol/L)	−0.083	0.570	0.047	0.703
TGL (mmol/L)	0.061	0.675	0.009	0.941
HDL (mmol/L)	−0.175	0.228	0.073	0.556
LDL (mmol/L)	−0.078	0.599	0.020	0.872
VLDL (mmol/L)	0.063	0.667	0.008	0.952
C peptide (pg/mL)	−0.014	0.927	−0.049	0.696
Serum Creatinine (mg/L)	0.291	0.043	−0.049	0.696
eGFR (mL/min/1.73 m^2^)	−0.225	0.123	0.018	0.883
BUN	0.198	0.174	−0.069	0.581
Albumin (mcg/L)	−0.274	0.057	−0.246	0.045
Insulin (mU/L)	−0.013	0.931	0.131	0.291
ACR (mg/g)	0.065	0.659	0.177	0.152
Urine Creatinine (mg/day)	−0.002	0.991	−0.014	0.909
Microalbumin (mg/day)	0.057	0.697	0.353	0.003
Ang2 (ng/mL)	0.197	0.175	0.093	0.455
ANGPTL3 (ng/mL)	0.235	0.104	0.257	0.036
ANGPTL4 (ng/mL)	0.138	0.343	−0.002	0.986
ANGPTL8 (ng/mL)	0.214	0.139	0.089	0.472

r, Pearson’s correlation coefficient with significance at *p* < 0.05.

**Table 3 biomedicines-12-00949-t003:** Pearson’s correlation analysis for Ang2 in study groups T2D and DN.

	Ang2			
Parameters	T2D		DN	
	r	*p*	r	*p*
Age (years)	0.277	0.054	0.135	0.277
BMI (kg/m^2^)	0.240	0.096	0.070	0.580
SBP (mmHg)	0.013	0.932	−0.280	0.024
DBP (mmHg)	−0.211	0.150	−0.209	0.095
Fasting Glucose (mmol/L)	0.203	0.162	0.100	0.419
HbA1C (%)	−0.024	0.871	0.028	0.820
T. Chol (mmol/L)	−0.204	0.159	−0.007	0.955
TGL (mmol/L)	0.063	0.669	0.196	0.113
HDL (mmol/L)	−0.176	0.227	−0.049	0.692
LDL (mmol/L)	−0.221	0.132	−0.099	0.428
VLDL (mmol/L)	0.063	0.666	0.195	0.114
C peptide (pg/mL)	0.032	0.827	−0.047	0.706
Serum Creatinine (mg/L)	0.300	0.036	0.215	0.081
eGFR (mL/min/1.73 m^2^)	−0.307	0.034	−0.282	0.021
BUN	0.320	0.025	0.236	0.054
Albumin(mcg/L)	−0.220	0.129	−0.189	0.126
Insulin (mU/L)	−0.060	0.683	−0.017	0.889
ACR (mg/g)	0.157	0.281	0.244	0.047
Urine Creatinine (mg/day)	−0.160	0.272	−0.157	0.204
Microalbumin (mg/day)	−0.040	0.783	0.092	0.461
Ang1 (ng/mL)	0.197	0.175	0.093	0.455
ANGPTL3 (ng/mL)	−0.123	0.401	0.190	0.123
ANGPTL4 (ng/mL)	0.555	0.001	0.541	0.001
ANGPTL8 (ng/mL)	0.020	0.889	0.410	0.001

r, Pearson’s correlation coefficient with significance at *p* < 0.05.

**Table 4 biomedicines-12-00949-t004:** Multiple regression analysis to identify parameters associated with Ang2. Adjusted for age, gender, and BMI.

Parameters	T2D		DN	
	β Coefficient	*p* Value	β Coefficient	*p* Value
SBP	−0.047	0.706	−0.273	0.012
ACR	0.078	0.535	0.345	0.002
ANGPTL4	0.552	<0.001	0.082	0.614
ANGPTL8	−0.099	0.438	0.424	<0.001

## Data Availability

The data presented in this study are available on request from the corresponding author.

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
