# Peer review of "Angiopoietin-2 and Angiopoietin-like Proteins with a Prospective Role in Predicting Diabetic Nephropathy"

_biomedicines, 2024, doi:10.3390/biomedicines12050949_

Round 1

Reviewer 1 Report

Comments and Suggestions for Authors

Manuscript entitled “Angiopoietin-2 and angiopoietin-like proteins with a prospective role in predicting diabetic nephropathy”

Introduction, aims of the study: more informative aim required to match the parameters included in the study

Figure 1: ANGPLTL 3 is not present as figure

Table 1: DN group also is with T2D. This should be indicated in the table

Table 1: Are the results for Ang1, Ang2, ANGPLT 4 and 8 the same as in Figure 1? If yes, they there is a repeated presentation of the results. They should appear in one representation form.

Table 4 – What is “beta”?

Figure 4: The orange line for ANGPLT8 in “B” does not match the red line in “C”.

Author Response

Figure 1: ANGPLTL 3 is not present as figure

We appreciate the reviewer’s point, accordingly a panel showing ANGPTL3 data is added to Figure1.

Table 1: DN group also is with T2D. This should be indicated in the table.

We appreciate the reviewer’s comment. DN stands for diabetic nephropathy i.e. people with T2D and nephropathy. This is now indicated in the table title and footnote (highlighted).

Table 1: Are the results for Ang1, Ang2, ANGPLT 4 and 8 the same as in Figure 1? If yes, they there is a repeated presentation of the results. They should appear in one representation form.

We appreciate the reviewer’s concern. True, Figure 1 plots reflect data from Table 1, which is highlighted in text line 148-157. As suggested by the reviewer, we have removed their details from table 1.

Table 4 – What is “beta”?

Beta in Table 4 is the regression coefficient. In the regression analysis conducted, beta (β) coefficients were estimated to assess the relationship between the independent variables and the dependent variable. Beta represents the change in the dependent variable for a one-unit change in the independent variable, while controlling for other variables included in the model.

 Figure 4: The orange line for ANGPLT8 in “B” does not match the red line in “C”.

We appreciate the reviewer’s comment. I believe the reviewer is referring to Figure 3B, and true there was a mistake in the figure that this is now corrected. A modified version of the plot is included in page 9 of the manuscript.

Reviewer 2 Report

Comments and Suggestions for Authors

Major points

1. In figure 1, the authors should provide the control group to compare circulating levels of Ang1, Ang2, ANGPTL4 and ANGPTL8 in people with T2D and DN.

2. The authors should provide a more detailed explanation of the mechanisms underlying the relationships between Ang2, ANGPTL4, and ANGPTL8 in the context of diabetic nephropathy.

3. The authors should adjust for the significant valuables for multiple regression analysis.

4. The authors should provide your study limitations in the discussion part.

5. The authors should provide the results of sequential levels of significant markers to establish a cause-and-effect relationship and track disease progression.

Minor points

1. In the abstract, DN should be provided as a full term.

Author Response

1. In figure 1, the authors should provide the control group to compare circulating levels of Ang1, Ang2, ANGPTL4 and ANGPTL8 in people with T2D and DN.

We appreciate the reviewer’s concern, and to answer this request we have included supplementary figure S.2 (Figure S2 in lines 511-518, pages 14-15) showing plots of all biomolecules (i.e. Ang1, Ang2, ANGPTL3, ANGPTL4, ANGPTL8) in healthy control, people with T2D and people with DN.

2. The authors should provide a more detailed explanation of the mechanisms underlying the relationships between Ang2, ANGPTL4, and ANGPTL8 in the context of diabetic nephropathy.

We appreciate the reviewer’s comment, however a detailed explanation of the mechanism connecting these biomolecules is what we are striving to achieve. In this report we are highlighting our finding, which is mainly an observation from a population-based study and trying to connect this to previously reported findings from diabetic nephropathy focused literature to reach a reasonable conclusion. Unfortunately, the nature of this study does not allow for reaching a mechanistic understanding of the relationship connecting these molecules with DN. However, we are hopeful that this study and others will be the setting stones for future studies that shall reflect a better image of the mechanistic relationship between these molecules and DN. 

We have added the following sentence to the limitation paragraph, and it reads as follows (in lines 228-330, page 13:

“Future research should include prognosis and mechanistic studies to confirm the correlation between Ang2 and ANGPTL(s) in DN. This would also clarify the cause-effect link between DN and these proteins.”

3. The authors should adjust for the significant valuables for multiple regression analysis.

We appreciate the reviewer comments. In this study multiple regression analysis was adjusted for age, gender and body mass index (BMI).

4. The authors should provide your study limitations in the discussion part.

As suggested by the reviewer, we have edited the last paragraph of the discussion to highlight the study limitations mainly its cross-sectional nature and it reads as follows (lines 325-330, page 13):

 “The main limitation of this study is the cross-sectional design of the study, which hindered the ability to determine the biological significance of these proteins and their potential involvement in the pathophysiology of DN and sequentially establish causality. Future research should include prognosis and mechanistic studies to confirm the correlation between Ang2 and ANGPTL(s) in DN. This would also clarify the cause-effect link between DN and these proteins.” 

5. The authors should provide the results of sequential levels of significant markers to establish a cause-and-effect relationship and track disease progression.

We appreciate the review comments. However, due to the cross-sectional nature of our study we cannot provide such data. We have added this as a limitation for this study and it reads as follows (lines 325-330, page 13):

“The main limitation of this study is the cross-sectional design of the study, which hindered the ability to determine the biological significance of these proteins and their potential involvement in the pathophysiology of DN and sequentially establish causality. Future research should include prognosis and mechanistic studies to confirm the correlation between Ang2 and ANGPTL(s) in DN. This would also clarify the cause-effect link between DN and these proteins.” 

Minor points

In the abstract, DN should be provided as a full term.

Full term “diabetic nephropathy” is added to the abstract.

Reviewer 3 Report

Comments and Suggestions for Authors

This manuscript clearly introduced epidemiology of diabetes and diabetic nephropathy. The pathogenic roles of angiopoietins are also well written. The methodology is clear. The main conclusion drawn by the authors are there is positive correlation between the elevation of Ang2 and increased levels of ANGPTL8 and ANGPTL4 in diabetic patients with DN, and the combination of Ang2 and ANGPTL8 could be a predictive tool for nephropathy in patients with T2D.

Here, I have a few recommendations for the author:

1. The biological functions of ANGPTLs need to be introduced and why only ANGPTL3, 4, 8 were chosen for this study.

2. Through this manuscript, the authors need to clearly state the two study groups (T2D without DN vs T2D with DN).

3. At the beginning of the abstract, the authors stated that angiopoietins are crucial for healthy functional endothelium, but also stated that angiopoietin 2 compromised endothelial integrity. This needs re-written.

4. The author concluded that Ang2 and ANGPTL8 could be risk predictors. However, is it possible that their levels are raised in response to renal injury as protective factor? Maybe, the authors need to consider compare the data between patients with mild DN to patients with severe DN.

5. The authors need to change 'accuracy' in page 3 line 120, 122 to more precise term for AUC purpose. Is'diagnositc strength' better?

6. The author need to specify the ethnic groups of the participants rather than using a general term 'Kuwaiti population'.

7. The author need to clarify what they mean by 'persistent' in page 3 line 132, 134.

8. For consistency, the author need to present ANGPTL3 data in Figure 1.

9. A lot of clinical parameters have been chosen for this study. The author could briefly explain why they were chosen.

10. For Figure 3, the AUC for ANGPTL8 is larger than that of Ang2 and ANGPTL8 combination, wouldn't this lead to a different conclusion?

11. Based on the above comments, the authors may need to expand the last paragraph of Discussion.

Comments on the Quality of English Language

Generally well written. A few minor changes are needed.

Author Response

1. The biological functions of ANGPTLs need to be introduced and why only ANGPTL3, 4, 8 were chosen for this study.

We appreciate the reviewer’s suggestion. We have added the following paragraph on ANGPTLs (lines 68-77, page2) and it reads as follows:

“ANGPTLs, particularly ANGPTL3, ANGPTL4, and ANGPTL8, play pivotal roles in regulating lipid metabolism and energy homeostasis, with emerging implications for diabetic nephropathy. ANGPTL3 and ANGPTL4 are known inhibitors of lipoprotein lipase (LPL), a key enzyme involved in triglyceride metabolism. By inhibiting LPL activity, these proteins modulate lipid clearance from the circulation, resulting in alterations in plasma lipid levels. ANGPTL8, often acting in conjunction with ANGPTL3, also regulates LPL activity and lipid metabolism, albeit with some distinct roles in glucose homeostasis and insulin sensitivity [14]. Dysregulation of these ANGPTLs has been implicated in the pathogenesis of metabolic disorders, including obesity, dyslipidemia, and type 2 diabetes mellitus (T2DM), all of which are risk factors for diabetic nephropathy [17].”

This report is a continuation of our interest and work on ANGPTL proteins, whereby in a previous publication entitled “ANGPTL4: A Predictive Marker for Diabetic Nephropathy, 2019” we have reported our finding with regard to ANGPTL4. Additionally, there are a number of publications reporting ANGPTL8 under the context of diabetes. All this justifies and explains our current choice.  

2. Through this manuscript, the authors need to clearly state the two study groups (T2D without DN vs T2D with DN).

We appreciate the reviewer’s concern, therefore we have included a sentence in the Materials and methods, 2.1 study population section to clearly state that T2D refers to people with diabetes, while DN refers to people with diabetes and nephropathy (line 94-95, page 2).

3. At the beginning of the abstract, the authors stated that angiopoietins are crucial for healthy functional endothelium, but also stated that angiopoietin 2 compromised endothelial integrity. This needs re-written.

We appreciate the reviewer’s comment, however Ang1 and Ang2 are critical proteins but have opposing biological functions. According to literature while Ang1 promotes endothelial cell survival and stabilization, Ang2 is a natural antagonist of Ang. Thus, Ang2 assists endothelial cell migration/proliferation with VEGF while it promotes vessel regression in the absence of VEGF. So, it is the selective elevation in levels of Ang2 and VEGF that seems to promote neovascularization and endothelial abnormalities that might contribute to microvascular pathophysiology.

4. The author concluded that Ang2 and ANGPTL8 could be risk predictors. However, is it possible that their levels are raised in response to renal injury as protective factor? Maybe, the authors need to consider comparing the data between patients with mild DN to patients with severe DN.

Following reviewer suggestion, we have stratified our study groups including DN cases based on creatinine levels into normal (Male <=119.3; Female <=91.9) and high (Male >119.3; Female >91.9). Serum levels for both Ang2 and ANGPTL8 seem to increase in both subgroups. However, their increased levels were more pronounced in the high creatinine subgroup (Supplementary table 2). Moreover, within the normal creatinine subgroup increased levels of both Ang2 and ANGPTL8 was evident in the DN group compared to the T2DM one, thus supporting the utility of both markers for early diagnosis of DN.

We have added the following paragraph on the result section in line 2108-217, page 8:

“We have also stratified our study groups including DN cases based on creatinine levels into normal (Male <=119.3; Female <=91.9) and high (Male >119.3; Female >91.9). Serum levels for both Ang2 and ANGPTL8 seem to increase in both subgroups. However, their increased levels were more pronounced in the high creatinine subgroup (Supplementary table 2). Moreover, within the normal creatinine subgroup increased levels of both Ang2 and ANGPTL8 was evident in the DN group compared to the T2DM one, thus supporting the utility of both markers for early diagnosis of DN.”

Supplementary table 1. Circulatory levels of ANGPTL8 and Ang2 in the T2D and DN groups were assessed in normal and high serum creatine stratification.

Serum Creatine-Normal

Serum Creatine-High

GROUP

VARIABLES

N

Mean ± SEM

N

Mean ± SEM

P value

T2DM

ANGPTL8

25

2372.93 ± 299.85

9

3954.19 ± 975.67

0.154

Ang2

25

3510.49 ± 385.95

9

4186.29 ± 1010.44

0.546

DN

ANGPTL8

23

3801.59 ± 603.94

41

4011.72 ± 409.64

0.775

Ang2

23

3985.57 ± 604.65

41

4616.07 ± 402.44

0.389

5. The authors need to change 'accuracy' in page 3 line 120, 122 to more precise term for AUC purpose. Is'diagnositc strength' better?

We appreciate the reviewer’s comment, the word ‘accuracy’ is replaced with diagnostic strength as advised.  

6. The author need to specify the ethnic groups of the participants rather than using a general term 'Kuwaiti population'.

We appreciate the reviewer’s comment. The sentence is modified to reflect the ethnicity of participants, that is Arabs from the population of the state of Kuwait.

7. The author need to clarify what they mean by 'persistent' in page 3 line 132, 134.

We appreciate the reviewer’s comment. The word ‘persistent’ has no scientific value in this text and for this it got removed.

8. For consistency, the author need to present ANGPTL3 data in Figure 1.

We appreciate the reviewer’s comment. A plot showing ANGPTL3 data is included in Figure 1 as panel ‘C’.  

9. A lot of clinical parameters have been chosen for this study. The author could briefly explain why they were chosen.

We appreciate the reviewer’s concern, but all the listed clinical data in Table 1 are of direct relevance to both type 2 diabetes and diabetic nephropathy. The purpose of listing these parameters is to give an anthropometric and clinical description of the participating population.   

Below are some descriptions on their relevance to diabetic nephropathy:

Blood Pressure (SBP and DBP): Hypertension is a well-established risk factor for diabetic nephropathy and is closely monitored in patients with diabetes.

Anthropometric Measures (Height and Weight): Body mass index (BMI) is associated with diabetic nephropathy risk, and height and weight measurements are essential for calculating BMI.

Glucose Metabolism (Fasting Glucose and HbA1C): Elevated fasting glucose and HbA1C levels are indicative of poor glycemic control, which is a key contributor to the development and progression of diabetic nephropathy.

Lipid Profile (TChol, TG, HDL-C, LDL-C, VLDL): Dyslipidemia, characterized by abnormalities in cholesterol and triglyceride levels, is commonly observed in patients with diabetic nephropathy and is associated with adverse renal outcomes.

C Peptide: C-peptide levels reflect endogenous insulin secretion and can provide insights into pancreatic beta-cell function, which is impaired in diabetes.

Renal Function Markers (Serum Creatinine, eGFR, BUN): Markers of renal function, such as serum creatinine, estimated glomerular filtration rate (eGFR), and blood urea nitrogen (BUN), are essential for diagnosing and monitoring diabetic nephropathy.

Urinary Markers (ACR, Urine Creatinine, Microalbumin): Albumin-to-creatinine ratio (ACR), urine creatinine, and microalbuminuria are important indicators of renal damage and are commonly used for early detection and monitoring of diabetic nephropathy.

Insulin: Insulin levels provide information about insulin resistance and beta-cell function, which are implicated in the pathogenesis of diabetic nephropathy.

These variables collectively offer a comprehensive assessment of key physiological and biochemical parameters relevant to the development and progression of diabetic nephropathy, enabling a thorough investigation of its pathophysiology and clinical manifestations in the manuscript.

10. For Figure 3, the AUC for ANGPTL8 is larger than that of Ang2 and ANGPTL8 combination, wouldn't this lead to a different conclusion?

We appreciate the reviewer’s concern. We have added the following sentence to the result section (lines 229-231, page 9):

“This finding indicates that ANGPTL8 may possess superior diagnostic accuracy compared to Ang2 and the combined biomarkers in distinguishing diabetic nephropathy from other conditions or outcomes. “

11. Based on the above comments, the authors may need to expand the last paragraph of Discussion.

We have expanded on the last paragraph of discussion, and it reads as follows:

“On the other hand, the larger AUC for ANGPTL8 underscores its potential as a robust biomarker for diabetic nephropathy, offering promise for improved diagnostic precision and patient stratification in clinical practice. Finally, the ROC analysis of the amalgamation of Ang2 and ANGPTL8 accentuates the significance of these proteins as prospective indicators for the progression into a state of nephropathy in people with T2D. Future research is needed to explain the relationship between Ang2 and ANGPTLs, particularly ANGPTL8 and ANGPTL4, as well as the mechanism by which they contribute to diabetic microvascular problems.”

Comments on the Quality of English Language

Generally well written. A few minor changes are needed.

Round 2

Reviewer 2 Report

Comments and Suggestions for Authors

The authors responded fully to the reviewers' comments. I fully understands the limitations of the study.